# Breaking Depth Estimation Models with Semantic Adversarial Attacks

Change in camera's viewpoint

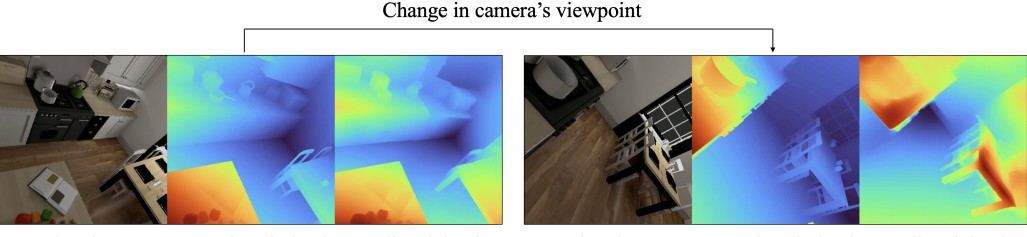

Rendered Image    Ground truth depth    Predicted depth       Rendered Image    Ground truth depth    Predicted depth

Figure 1: Starting from a view of a 3D scene where a depth estimation model works well (left), our method changes the camera's viewpoint so that the model's prediction is no longer accurate.

## Abstract

Monocular depth estimation models have advanced significantly in recent years, where it seems as if they can provide accurate depth information for any arbitrary scene. In this work, we develop a framework to see if this indeed is true by stress-testing them in different indoor environments. Specifically, our goal is to study how robust various models are to changes in camera viewpoint. Rather than conducting an exhaustive search over all possible viewpoints in a scene, we employ adversarial attacks leveraging a differentiable rendering framework applied to 3D assets. By initializing from a given camera position, we optimize the camera's rotation and translation parameters through backpropagation to update prediction errors. To ensure meaningful failure cases, we implement strategies that prevent trivial adversarial shortcuts. To make all of this possible, we also construct a dataset comprising of complex, efficiently renderable 3D assets, enabling rigorous evaluation of four recently published depth estimation models. The key insight from our experiments is that all of those models, including the very recent state-of-the-art, fail on the adversarial viewpoints discovered through our framework, i.e., their predictions deviate significantly from the ground-truth depth. Our work establishes a new robustness benchmark for monocular depth estimation task.

## 1 Introduction

Predicting depth from an RGB image is a crucial task for many computer vision applications. The initial algorithms tackled this problem for localized settings; neural networks were trained to work on specific domains, e.g., only for self-driving car datasets Geiger et al. (2013) or only for indoor scenesSilberman et al. (2012). This paradigm changed for monocular depth estimation (MDE) task through methods that enable training the same model on diverse datasets of much bigger scale Ranftl et al. (2020). Consequently, we now have a suite of MDE models that seem to work pretty well on images in the wild Yang et al. (2024a;b).

Given how powerful these models seem to be, it is natural to ask if they fail at all. In this work, we study the robustness of MDE models to semantic properties of the image; specifically, to camera's position. Imagine if a camera had the liberty to move around anywhere in a 3D scene, capturing images from all sorts of viewpoints; getting close to objects, flying above them looking top down, going into a corner of the room etc. Would these powerful models remain powerful for images

captured from all of these viewpoints? Or can they fail at certain viewpoints? Questions such as these are important because such scenarios can naturally arise in the real world when, e.g., a robot moves through an environment estimating depth as it goes. The naive way to study such a property could be to exhaustively sample images from many viewpoints in the scene from a real world, obtain their ground-truth depth using some appropriate device (e.g., Microsoft Kinect), and measure dissimilarity with the model's prediction. However, finding failure cases this way will be very time consuming. A better alternative could be to do this study on rendered images from 3D assets where the rendering tool can automatically compute the ground-truth depth. Even in this case, one will need to search over many viewpoints to find ones where the model fails.

Our algorithmic contribution is to make this search process *directed* by updating the camera position so that it moves to locations where the model fails. We do this with the help of a differentiable renderer. Given a 3D asset, a starting camera position parameterized by rotation and translation matrix, we render the RGB image and obtain its ground-truth depth. Both are computed differentiably with respect to the camera's parameters. Feeding the image into the MDE model, we compute the loss between the predicted and ground-truth depth and backpropagate the gradients to update the rotation and translation parameters to increase the loss. We also apply a penalty so that no shortcuts are taken to increase the loss, e.g., by going inside an object, or by going outside the 3D mesh, both of which will render meaningless images. Performing this gradient ascent over a series of steps helps us find the meaningful failure modes for many different models across multiple 3D assets.

This whole process relies not only on the availability of 3D assets that can be rendered not just differentiably, but also quickly, since a new scene has to be rendered after each update. Since no such dataset of complex scenes exists to the best of our knowledge, we collect nine public 3D assets, and convert them into a format usable with PyTorch. Using this dataset, we study four major depth estimation models that have been proposed in the last five years, including state of the art Depth Anything v1/v2. We find that all models fail in providing accurate depth prediction for adversarial viewpoints. The objective of our work is less to study the precise nature of scene factors that break a model (e.g., specific viewpoint) but more to develop a benchmark to stress-test MDE models to see how robust they can be in certain environments. Overall, our contributions are the following:

- We propose and formulate a new task of finding failure cases of modern depth estimation algorithms with respect to camera's external parameters.

- To to that, we propose 3D assets that can be rendered differentiably and quickly. We plan to release the assets.

- Using these assets, we create an adversarial attack framework that optimizes camera's parameters, without taking shortcuts, to maximize depth prediction errors.

- We study four state-of-the-art monocular depth estimation algorithms, and find that our toolbox can discover failure cases for all of them.

## 2 RELATED WORK

**Depth Estimation.** The task of depth estimation involves predicting the distance of objects present in an image from the camera. Traditionally, this was done in a stereo setting, using left and right camera images Scharstein & Szeliski (2003); Kusupati et al. (2020). More recently with the proliferation of deep learning systems, it is becoming possible to do the same task using just a single image, i.e., monocular depth estimation. Within this line of work, there are two broad categories; models which predict metric depth Yin et al. (2023); Bhat et al. (2023), and those which predict relative depth Lee & Kim (2019); Chen et al. (2016). For the latter category, MiDaS Ranftl et al. (2020) made a crucial breakthrough by developing a scale and shift invariant objective function that allows training the same model across multiple types of datasets. This key idea was further used to enable using huge amounts of unlabeled images to create very powerful depth estimation model - DepthAnything Yang et al. (2024a). A follow up work made use of the higher quality ground-truth depth information from synthetic datasets to propose a stronger model DepthAnything V2 Yang et al. (2024b). More recently, there has been an attempt to re-purpose image generative models into depth estimators Ke et al. (2024). All of these models that have come out in the last few years seem capable to be deployed for all kinds of scenes in the wild. However, it is not clear how these new models can fail. In this work, we study whether they do struggle with camera viewpoint changes.

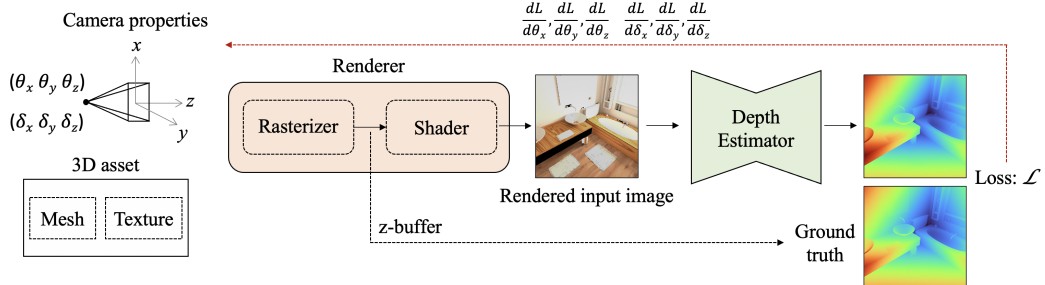

Figure 2: Our overall pipeline for finding failure modes. Given a 3D asset and a camera initialized with some rotation $\theta = \{\theta_x, \theta_y, \theta_z\}$ and translation parameters $\delta = \{\delta_x, \delta_y, \delta_z\}$, PyTorch3D renders the image $\mathcal{I}_{\theta,\delta}$ in a differentiable way. We obtain ground-truth depth $\mathcal{D}_{\theta,\delta}$ through an intermediate step producing z-buffer values. The rendered image goes to an arbitrary model $\phi$ which predicts depth $\hat{\mathcal{D}}_{\theta,\delta}$. Camera parameters are updated to increase the discrepancy between $\mathcal{D}_{\theta,\delta}$ and $\hat{\mathcal{D}}_{\theta,\delta}$.

**Adversarial attack** Neural networks have been shown to be susceptible to adversarial attacks, where a purposefully crafted imperceptible noise added to the input completely changes a model's output. This was first studied for image classification Goodfellow et al. (2015); Kurakin et al. (2017), where the noise was added onto the pixels of the image. This was extended to other computer vision tasks, like semantic segmentation Xie et al. (2017); Arnab et al. (2018), object detection Liu et al. (2019), depth estimation Zhang et al. (2020); Zheng et al. (2024). There has also been a line of work which goes beyond pixel level attacks to more semantic attacks; e.g., fooling classifiers by changing lighting condition, surface normals Zeng et al. (2019); Jain et al. (2020), by imperceptibly changing the mesh Xiao et al. (2019), by changing camera viewpoint Dong et al. (2022). Since these works optimize factor(s) in the semantic feature space, they often use some form of differentiable rendering Ravi et al. (2020); Li et al. (2018); Kato et al. (2018); Jakob et al. (2022) or Nerf-based techniques Mildenhall et al. (2020). Any adversarial attack is essentially trying to optimize some property to progressively deviate a model's prediction from the ground-truth. A key similarity in all of the works is that this ground-truth (e.g., class label, ground-truth depth) is assumed to be constant throughout the optimization process. Our objective in this work differs in this key aspect; since our goal is to change the camera viewpoint, the ground-truth depth at each step also changes. We make use of the differentiable rendering tool to handle this.

## 3 METHOD

An RGB image captured with a camera has a corresponding ground-truth depth map whose each pixel indicates the distance of the corresponding object from the camera. The task of a monocular depth estimation model is to process the RGB image and estimate this ground-truth depth. Our goal is to find cases where this estimation fails because of changes in the camera viewpoint. That is, starting from a location of the camera where a MDE model predicts depth accurately, find a location within the same scene where it does not do a good job.

This problem formulation has two requirements. First, we should be able to change camera location and render a new image of the same scene. Second, for each camera location, we also need access to ground-truth depth information of the corresponding RGB image. Consequently, we work with 3D-assets of indoor scenes. Please see the appendix why 3DGS/NeRF based techniques cannot give us accurate depth of the scenes. We first give an overview of how a camera, the factor that we want study, is parameterized for 3D computer vision tasks. Following that, we explain how its parameterization, along with 3D assets, can be used to find the camera viewpoints where a model struggles.

### 3.1 PRELIMINARY NOTES

A camera has many properties. Of particular interest to us in this work are those which define its position and orientation in the 3D world. The orientation of the camera can be described using the three axis angles - $\theta = \{\theta_x, \theta_y, \theta_z\}$ - in the world coordinate system. The axis angles can be

converted into the more common form of a $3 \times 3$ rotation matrix. All valid rotation matrices can be represented using $\theta_{rot}$. The position of the camera can be described using three translation parameters - $\delta = \{\delta_x, \delta_y, \delta_z\}$ which define the $3 \times 1$ translation vector. Collectively, these two form the extrinsic matrix, which describes how to transform the points from the world coordinate system to camera coordinates. Henceforth, we refer to $\theta_{rot}$ and $\delta_{shift}$ as $\theta$ and $\delta$ respectively for simplicity. These six parameters can represent a camera in any location looking at any direction.

Given this transformation, the image gets rendered through a sequence of steps (e.g., projection, rasterization). One of the intermediate steps of the rendering process also computes the z-component of the Euclidean distance of a point in a scene from the camera, a.k.a, depth map. This will serve as our ground-truth depth. Both spatial signals, the rendered RGB image and its ground-truth depth, are functions of the rotation and translation properties of the camera - $\mathcal{I}_{\theta,\delta}$ and $\mathcal{D}_{\theta,\delta}$. $\mathcal{I}_{\theta,\delta} \in R^{H \times W \times 3}$ and $\mathcal{D}_{\theta,\delta} \in R^{H \times W}$, where $H, W$ are height and width of the rendered image. We use a differentiable renderer (PyTorch3D) to compute both of these values, which will be critical, as we later explain, to update $\theta$ and $\delta$ given a model's accuracy.

## 3.2 FORWARD PASS

Given a 3D asset, we first initialize $\theta$ and $\delta$ to some values so that (i) the rendered image is some reasonable, commonplace view of the scene, and (ii) a depth estimation model $\phi$ performs reasonably well on that image. The image is then passed into a MDE model, $\phi$, that is of interest to us (e.g., DepthAnything Yang et al. (2024a)). The predicted depth, which is also a function of the same camera parameters, is denoted as $\phi(\mathcal{I}_{\theta,\delta}) = \hat{\mathcal{D}}_{\theta,\delta}$. Both, the ground truth and predicted depth maps are normalized to have values in $[0, 1]$. The models that we are interested in studying are big and powerful enough so that for such commonplace views, $\hat{\mathcal{D}}_{\theta,\delta} \approx \mathcal{D}_{\theta,\delta}$ typically. We want to update $\theta$ and $\delta$ to some values so that $\hat{\mathcal{D}}_{\theta,\delta} \neq \mathcal{D}_{\theta,\delta}$.

To do that, we first need a metric to determine how similar or dissimilar two depth maps are. Measuring distance on pixel level doesn't always correspond well to humans' judgments of similarity. A common example is an image and its blurrier version; humans can recognize their differences, something that pixel-wise L2 distance, which is somewhat low for the pair, does not show. This was studied extensively for RGB images in Zhang et al. (2018), and the authors found perceptual loss Johnson et al. (2016), which measures the same L1 distance in a more semantic feature space, to better correspond to human judgments. On account of similar requirements, we employ the same loss for our task. We pass both $\mathcal{D}_{\theta,\delta}$ and $\hat{\mathcal{D}}_{\theta,\delta}$ into a VGG-16 Simonyan & Zisserman (2015) model pre-trained on ImageNet dataset Deng et al. (2009), which we denote as $\psi$. Across a pre-defined set of layers in $\psi$, $L = \{l_1, l_2, ...\}$, we extract features for $\mathcal{D}_{\theta,\delta}$ and $\hat{\mathcal{D}}_{\theta,\delta}$ and compute averaged L1 difference. The process can be summarized formally as following:

$$\mathcal{L}_{depth}(\mathcal{D}_{\theta,\delta}, \hat{\mathcal{D}}_{\theta,\delta}) = \sum_{j \in L} \frac{1}{C_j H_j W_j} |\psi_j(\mathcal{D}_{\theta,\delta}) - \psi_j(\hat{\mathcal{D}}_{\theta,\delta})| \tag{1}$$

where $\psi_j$ denotes output after $j^{th}$ layer, and $H_j, W_j, C_j$ denote its height, width, channels.

## 3.3 ADVERSARIAL ATTACK

Once $\mathcal{L}_{depth}$ is computed, our next goal is to maximize it. This bears similarity to the traditional adversarial attack for image classification models where noise is added to an image to increase the cross entropy loss in order to flip the label predicted by the model. However, there is a crucial difference. For image classification, the attack is $L_{\infty}$ norm bounded, restricting the added noise to remain imperceptible to the human eye. Because of this, the attack objective assumes the ground-truth label of the adversarial image to remain constant. In our case, we are interested in *exploring* the whole scene to find failure cases, because of which we do not impose any analogous imperceptibility bound on $\theta$ and $\delta$. However, because of that, each update $\theta \rightarrow \theta', \delta \rightarrow \delta'$ also changes the ground-truth depth ($\mathcal{D}_{\theta,\delta} \rightarrow \mathcal{D}_{\theta',\delta'}$). So, when we perform gradient *ascent* to maximize $\mathcal{L}_{depth}(\mathcal{D}_{\theta,\delta}, \hat{\mathcal{D}}_{\theta,\delta})$, we backpropagate the gradients through both $\mathcal{D}_{\theta,\delta}$ and $\hat{\mathcal{D}}_{\theta,\delta}$. The forward and backward passes are depicted in Fig. 2. Each step updates the six parameters ($\theta + \delta$) of the camera.

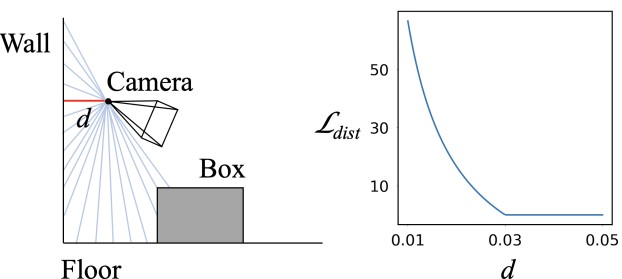

Figure 3: **Left.** A toy example where distance of the camera is computed from all nearby mesh surfaces (light blue lines). $d$ is set to be the shortest distance among them (red line). **Right.** Plot showing how the loss function $\mathcal{L}_{dist}$ varies as $d$ changes. Here, $d_{th}$ is set to be 0.03. There is penalty only when the camera comes very close to a surface ($d < d_{th}$).

### 3.4 PREVENTING SHORTCUTS

There are some problems with maximizing $\mathcal{L}_{depth}(\mathcal{D}_{\theta,\delta}, \hat{\mathcal{D}}_{\theta,\delta})$ without any constraints on $\theta$ and $\delta$. After being initialized at a reasonable place, successive updates can move the camera so that it either clips into an object (partially or completely) or goes outside the mesh altogether. Both of these cases will result in rendered image $\mathcal{I}_{\theta,\delta}$ being unnatural, on which failure of the MDE model $\phi$ will be meaningless. We penalize such shortcuts from being taken. The process is explained in Fig. 3(left). At each step, we compute the distance of the optical center of the camera to the closest mesh surface, denoted as $d$. When $d$ becomes less than a threshold, $d_{th}$, we introduce a penalty term inversely proportional to $d$. Formally, we define the penalty, $\mathcal{L}_{dist}$ as follows:

$$\mathcal{L}_{dist} = \begin{cases} \frac{1}{d} - \frac{1}{d_{th}}, & \text{if } d < d_{th} \\ 0, & \text{otherwise.} \end{cases} \tag{2}$$

We subtract a constant of $\frac{1}{d_{th}}$ so that the resulting curve of $\mathcal{L}_{dist}$ vs $d$, which is shown in Fig. 3(right), is relatively smooth at $d = d_{th}$. This penalty spikes up whenever the camera gets very close to any object (e.g., a table, wall, floor) in the 3D asset. Our final loss function is $\mathcal{L} = \max_{\theta,\delta} \mathcal{L}_{depth} + \min_{\theta,\delta} \mathcal{L}_{dist}$

### 3.5 VISUALIZING FAILURE MODES

We evaluate each MDE model on nine assets. For each asset/scene, we initialize the camera at ten different viewpoints which produce reasonable images of the scene, ensuring that in all cases the starting camera's position has $d > d_{th}$. We denote this set as `init-params` $= \{(\theta^0, \delta^0), (\theta^1, \delta^1), ...(\theta^{10}, \delta^{10})\}$. From each viewpoint, we run the optimization described above five times, each time for $N_{steps} = 400$ steps, and select the run where $\hat{\mathcal{D}}_{\theta,\delta}$ deviates most from $\mathcal{D}_{\theta,\delta}$ during the trajectory. We do multiple runs because each run is different due to some inherent stochasticity of the rendering process (please see appendix for details). For an arbitrary starting location $(\theta^i, \delta^i)$, the selected run is essentially a sequence of camera's trajectory which we denote as $\{(\theta^i_0, \delta^i_0), (\theta^i_1, \delta^i_1), ...(\theta^i_{400}, \delta^i_{400})\}$, where $(\theta^i_j, \delta^i_j)$ is the position of the camera after $j^{th}$ iteration after starting from $i^{th}$ viewpoint. Note that $(\theta^i, \delta^i) = (\theta^i_0, \delta^i_0)$.

We consider our whole setup of the differentiable rendering of the 3D assets + the adversarial attack as a toolbox for the users to diagnose a MDE model. The sequence of $\theta, \delta$ returned from the algorithm could be studied as a video of the camera's movement in the scene leading to its failure. Or we can sample $N_{adv} = 5$ positions $(\theta + \delta)$ from this trajectory where $\mathcal{L}_{depth}$ is highest, which can be considered as adversarial viewpoints.

## 4 3D ASSETS CREATION

The method described above is feasible only if there are 3D assets that can be rendered differentiably. So, there are three requirements from the assets - (i) they need to be depicting somewhat complex scenes, e.g., living room, since those are the scenarios where modern MDE models are deployed (assets of singular objects will therefore won't be of much help); (ii) the asset files need to be in a format compatible with PyTorch3D; (iii) they should be publicly accessible so that others can use our final toolbox, or build upon it. To the best of our knowledge, we could not find any existing repository

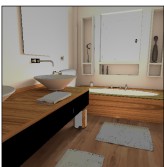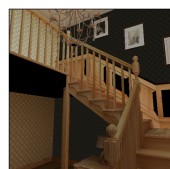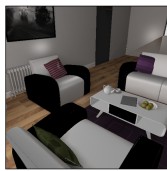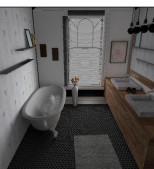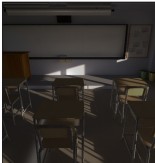

Figure 4: Images from the 6 (out of 9) 3D assets that we use in this work.

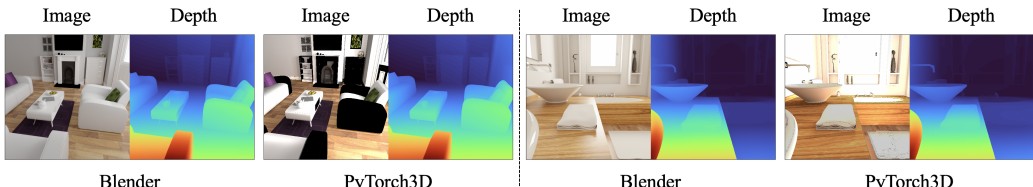

Figure 5: Two sample scenes (left and right halves) showing how the 3D asset looks after being rendered using Blender (left) and PyTorch3D (right). The depth maps obtained using Depth Anything-V2 Yang et al. (2024b) for both images are similar, despite the small quality drop in RGB image.

of 3D assets of complex scenes that can be rendered using PyTorch3D. So, we took nine publicly available 3D assets of complex scenes created for traditional renderers, e.g., Blender, and converted them into PyTorch3D compatible format. We show some samples in Fig. 4. Next, in Fig. 5, we show two sample conversions; on top are two images rendered using Blender, and on bottom are the same assets converted and rendered using PyTorch3D. The difference in rendering techniques (ray tracing for blender and rasterization for PyTorch3D) leads to some difference in the quality of the rendered images. We study if this difference is important later on in Sec. 5.1.

## 5 EXPERIMENTS

We first discuss the different MDE models that we study in this work. Then, we do a small study to confirm whether those MDE models' behavior on PyTorch3D rendered scenes is still comparable to the original Blender scenes. Finally, we discuss our experiments where we try to find adversarial viewpoints for those different MDE models.

**Models studied.** (i) MiDaS Ranftl et al. (2020), which was a seminal work enabling training models across multiple datasets. It has multiple variants based on model's size - small, medium and large, and we use the largest and strongest version in this work. (ii) ZoeDepth Bhat et al. (2023), which was proposed as a way to combine relative and metric depth estimation. We use the version finetuned on NYU Silberman et al. (2012) and KITTI Geiger et al. (2013) dataset. (iii) DepthAnythingV1 Yang et al. (2024a), which proposed using unlabeled images for better performance. (iv) DepthAnythingV2 Yang et al. (2024b), which was partly trained on synthetically rendered images (initial training was done on 500k synthetic images and then trained on 62 million real images).

| | | | | Asset ID | | | | | |
|------|------|------|------|------|------|------|------|------|
| 1 | 2 | 3 | 4 | 5 | 6 | 7 | 8 | 9 |
| 0.97 | 0.91 | 0.93 | 0.93 | 0.98 | 0.91 | 0.78 | 0.94 | 0.81 |

Table 1: From the same viewpoint in a 3D scene, we render images from Blender and PyTorch3D and obtain their respective depth maps using Depth Anything V2 Yang et al. (2024b). We measure their similarity using $\delta_1$ score , which can be from [0,1] ($\uparrow$ means more similar).

### 5.1 RELIABILITY OF PYTORCH3D RENDERINGS

The ultimate goal of the paper is to study how sensitive MDE models could be to uncommon viewpoints in the real world. But since we cannot algorithmically test a model in the real world,

we approximate the domain of its images through PyTorch3D renderings of scenes. However, as explained in Sec. 4, the (converted) assets rendered through PyTorch3D cannot completely match the quality of renderings produced through Blender. So, in this section, we investigate whether the depth prediction of a model on an RGB image is roughly the same whether the image is rendered using PyTorch3D or Blender.

Specifically, for each asset, we consider the 10 different viewpoints specified by `init-params`. For each such viewpoint, we generate image using PyTorch3D - $\mathcal{I}_{\theta,\delta}^{Pt}$, and Blender - $\mathcal{I}_{\theta,\delta}^{Bl}$. Then, we using Depth Anything V2 as $\phi$, we obtain two depth maps $\phi(\mathcal{I}_{\theta,\delta}^{Pt})$ and $\phi(\mathcal{I}_{\theta,\delta}^{Bl})$. We measure how similar these predicted maps are using threshold accuracy $\delta_1$, which measures the percentage of predicted pixels that differ from the ground-truth pixels by no more than 25% (higher is better). We do this across all the nine assets.

First, we show the qualitative results in Fig. 5, where we see that the depth maps predicted for $\mathcal{I}_{\theta,\delta}^{Pt}$ and $\mathcal{I}_{\theta,\delta}^{Bl}$ roughly look the same to the human eye for two different scenes. We include more qualitative results in the appendix. Next, we show the quantitative results in Table 1, where we see that across all the nine assets, $\delta_1$ score ($\uparrow$ means similar) is consistently high. The results indicate that, despite some quality difference, the performance of $\phi$ on our PyTorch3D renderings can be indicative of its behavior in higher quality renderings as well.

## 5.2 SEMANTIC ADVERSARIAL ATTACK

Now, using the same 3D assets developed for rendering using PyTorch3D, we perform the adversarial attack as described in Sec. 3.4. Our experimental setup is the following. For each 3D asset and each starting camera location in `init-params`, obtain top $N_{adv}$ viewpoints from the returned trajectory. We can measure how successful these adversarial viewpoints are in two ways. First is by qualitatively and quantitatively evaluating a model $\phi$'s performance on the images rendered from adversarial viewpoints. Second is by quantitatively studying how bad model's performance is *with respect to* the starting (benign) viewpoint. We do this across all the four models described above.

**Visualizing failure cases.** First, we show the qualitative results in Fig. 6. Each cell corresponds to an adversarial viewpoint, and is a triplet where the left, middle and right entities are rendered image, ground-truth depth and predicted depth respectively. The results for different models are in different rows. Our first takeaway is that every model breaks for certain viewpoints where it struggles to predict accurate depth. While the objective of our work is *not* to describe the precise conditions under which a particular model fails, we nonetheless observe certain patterns. Sometimes, the distribution of the predicted depth values is a bit off, where the close by objects are not predicted as close enough compared to ground-truth, e.g., row three, right cell. There are cases where a model predicts depth incorrectly for an object or for multiple objects, e.g., row 5, left cell. In other cases, the model fails to mimic the fine-grained textured nature of the ground-truth depth, e.g., row 1, right cell. Other times, a model fails when the rendered view of the scene contains an object or a part of the object apparently hanging in the air; e.g., row 4/6 right cell, Fig. 9 rightmost. Qualitative results for Depth Anything V2 are presented in the appendix. The final point to note is that while the manner in which failure happens is different, they all correspond to a high value of $\mathcal{L}_{depth}$, as we see next. This is because perceptual loss, which operates in a semantic feature space, captures and responds to many types of image dissimilarities.

**Quantitative results.** Next, we quantify these failure cases using two metrics; $\mathcal{L}_{depth}$ and $\delta_1$. While $\delta_1$ is a standard metric whose value (between 0 and 1; 1 means most similar) can indicate the degree of depth prediction failure, the same is not true for the value of $\mathcal{L}_{depth}$. So, to aid the reader in visualizing $\mathcal{L}_{depth}$ score, we show four degrees of depth prediction failure in Fig. 9. Each pair has the corresponding $\mathcal{L}_{depth}$ score at its top. From this, a general rule of thumb can be that $\mathcal{L}_{depth} > 0.8$ means that deviation of prediction is big enough to be considered a failure case.

In Table 2, we report these two scores achieved by the adversarial viewpoints. Each column corresponds to one of the nine assets, and each row corresponds to a different MDE model. Each score is averaged across the adversarial viewpoints obtained from all ten starting viewpoints from `init-params`. Again, the first take away is that all MDE models achieve a high $\mathcal{L}_{depth}$ ($> 0.9$) and low $\delta_1$ score ($< 0.33$). This means that predictions sufficiently deviate from ground-truth depth.

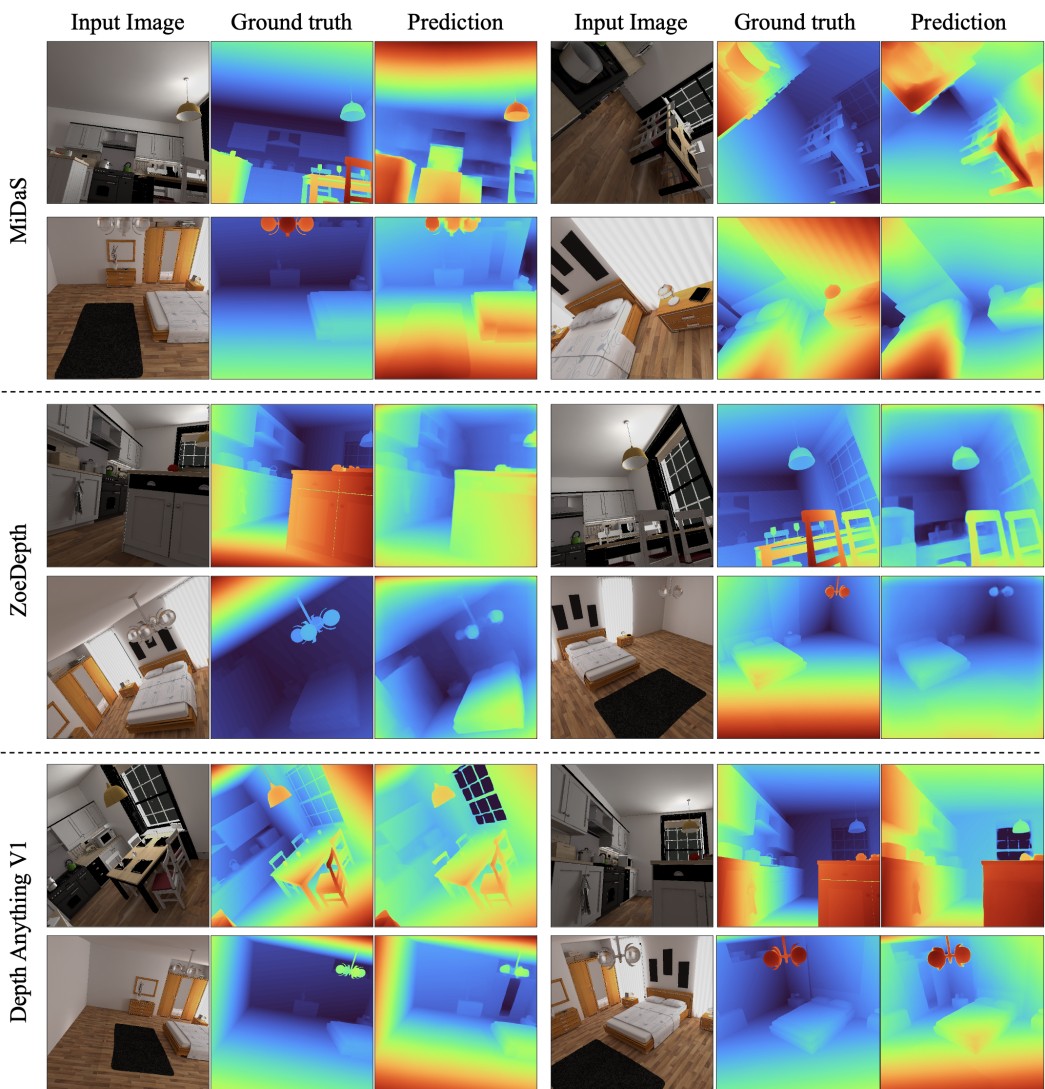

Figure 6: Adversarial viewpoints obtained for different models. Each triplet contains rendered image, ground-truth and predicted depth.

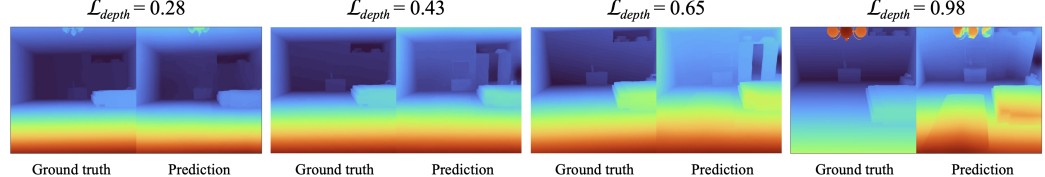

Figure 7: Predicted and ground-truth depth map pairs from four different camera viewpoints. The viewpoints are sampled from a trajectory during the optimization process, and the value of $\mathcal{L}_{depth}$ is increasing from left to right. The figure illustrates what different values of $\mathcal{L}_{depth}$ mean visually.

However, we also note that, generally speaking, more recent and powerful models like DepthAnything V1 and V2 (DA V1 and V2) fail less than their predecessors MiDaS and ZoeDepth. For example, the average $\delta_1$ score for DA V1/V2 (0.29) is $\sim 40\%$ higher than the score for the remaining two models (0.21). This is expected as the more recent models were built upon the earlier ones with the added advantage of a much bigger dataset. Among these, the adversarial attack is least successful for DA V2. The reason for this could be the way DA V2 was trained using synthetic data. Specifically,

| Model | Asset 1 | | Asset 2 | | Asset 3 | | Asset 4 | | Asset 5 | | Asset 6 | | Asset 7 | | Asset 8 | | Asset 9 | | Avg | |
|---|---|---|---|---|---|---|---|---|---|---|---|---|---|---|---|---|---|---|---|---|
| | $\mathcal{L}_{depth}$ | $\delta_1$ | $\mathcal{L}_{depth}$ | $\delta_1$ | $\mathcal{L}_{depth}$ | $\delta_1$ | $\mathcal{L}_{depth}$ | $\delta_1$ | $\mathcal{L}_{depth}$ | $\delta_1$ | $\mathcal{L}_{depth}$ | $\delta_1$ | $\mathcal{L}_{depth}$ | $\delta_1$ | $\mathcal{L}_{depth}$ | $\delta_1$ | $\mathcal{L}_{depth}$ | $\delta_1$ | $\mathcal{L}_{depth}$ | $\delta_1$ |
| MiDaS | 1.53 | 0.26 | 1.11 | 0.10 | 1.19 | 0.20 | 1.06 | 0.22 | 1.15 | 0.18 | 1.73 | 0.32 | 2.35 | 0.17 | 1.17 | 0.29 | 1.88 | 0.26 | 1.46 | 0.22 |
| ZoeDepth | 1.42 | 0.22 | 0.95 | 0.09 | 1.30 | 0.10 | 1.11 | 0.21 | 1.05 | 0.14 | 1.61 | 0.32 | 2.48 | 0.24 | 1.31 | 0.13 | 1.83 | 0.33 | 1.45 | 0.20 |
| Depth Anything V1 | 1.26 | 0.31 | 0.74 | 0.23 | 1.04 | 0.20 | 0.99 | 0.09 | 0.94 | 0.20 | 1.23 | 0.50 | 1.74 | 0.37 | 1.06 | 0.38 | 1.75 | 0.22 | 1.19 | 0.28 |
| Depth Anything V2 | 1.07 | 0.42 | 0.68 | 0.35 | 0.85 | 0.24 | 0.76 | 0.09 | 0.67 | 0.11 | 0.99 | 0.35 | 1.28 | 0.46 | 0.98 | 0.44 | 1.23 | 0.29 | 0.95 | 0.31 |

Table 2: $\mathcal{L}_{depth}$ and $\delta_1$ scores depicting discrepancy between depth predicted from adversarial viewpoints and the corresponding ground-truth. $\uparrow \mathcal{L}_{depth}$ and $\downarrow \delta_1$ denote more discrepancy.

| Model | Asset 1 | | Asset 2 | | Asset 3 | | Asset 4 | | Asset 5 | | Asset 6 | | Asset 7 | | Asset 8 | | Asset 9 | | Avg | |
|---|---|---|---|---|---|---|---|---|---|---|---|---|---|---|---|---|---|---|---|---|
| | $\mathcal{L}_{depth}$ | $\delta_1$ | $\mathcal{L}_{depth}$ | $\delta_1$ | $\mathcal{L}_{depth}$ | $\delta_1$ | $\mathcal{L}_{depth}$ | $\delta_1$ | $\mathcal{L}_{depth}$ | $\delta_1$ | $\mathcal{L}_{depth}$ | $\delta_1$ | $\mathcal{L}_{depth}$ | $\delta_1$ | $\mathcal{L}_{depth}$ | $\delta_1$ | $\mathcal{L}_{depth}$ | $\delta_1$ | $\mathcal{L}_{depth}$ | $\delta_1$ |
| MiDaS | 0.66 | 0.33 | 0.57 | 0.54 | 0.41 | 0.33 | 0.35 | 0.19 | 0.64 | 0.54 | 0.85 | 0.27 | 0.67 | 0.08 | 0.42 | 0.2 | 0.56 | 0.09 | 0.57 | 0.29 |
| ZoeDepth | 0.47 | 0.20 | 0.28 | 0.08 | 0.48 | 0.31 | 0.33 | 0.45 | 0.32 | 0.11 | 0.51 | 0.25 | 0.69 | 0.00 | 0.4 | 0.29 | 0.54 | 0.07 | 0.45 | 0.20 |
| Depth Anything V1 | 0.63 | 0.47 | 0.33 | 0.6 | 0.54 | 0.40 | 0.51 | 0.75 | 0.55 | 0.69 | 0.54 | 0.34 | 0.43 | 0.19 | 0.46 | 0.39 | 0.71 | 0.21 | 0.52 | 0.45 |
| Depth Anything V2 | 0.59 | 0.49 | 0.37 | 0.56 | 0.45 | 0.34 | 0.38 | 0.78 | 0.38 | 0.77 | 0.46 | 0.45 | 0.30 | 0.18 | 0.45 | 0.36 | 0.44 | 0.32 | 0.42 | 0.47 |

Table 3: Difference in $\mathcal{L}_{depth}$ score for adversarial and initial viewpoints. A positive value means $\mathcal{L}_{depth}$ increased for adversarial viewpoint. Similarly, we also show the negative of difference (for better readability) in $\delta_1$ scores for initial and adversarial viewpoints. A positive score indicates that $\delta_1$ score decreased for adversarial viewpoint, meaning deviating away from the ground-truth.

one of the synthetic datasets used during its training, Hypersim (Roberts et al., 2021), is created through realistic renderings of 3D assets, where from the same asset, multiple scenes are captured in a camera trajectory designed by an artist. Hence, we can expect DA V2 to be somewhat more robust to different camera viewpoints in our evaluation.

**Adversarial vs benign viewpoints.** So far, we saw that MDE models fail to produce accurate depth on the adversarial viewpoints returned from our toolbox. However, the scores for adversarial viewpoints can be even better understood in the context of the scores for more benign viewpoints. More specifically, we study how these scores look for images rendered from the initialized viewpoints (init-params). For each starting position in init-params, we compute $\mathcal{L}_{depth}^{adv} - \mathcal{L}_{depth}^{init}$. Similarly, we compute $\delta_1^{init} - \delta_1^{adv}$. We reverse the order of operands so that a positive difference value implies the same trend for $\mathcal{L}_{delta}$ and $\delta_1$. We create a *difference* Table 3, similar to Table 2, by averaging these differences across the ten viewpoints, for each asset, model combination. The first thing to notice is that *all* entries are positive, which means that predicted depth maps become less accurate for all assets and models, measured by both the metrics. The next worthwhile thing to notice is the degree to which depth maps become worse compared to the initial position. For all models, the average change in $\delta_1$ is more than $0.2$, and for DA V1/V2, it is more than $0.4$. The same story exists for $\mathcal{L}_{depth}$, where the average increase across all the models is $0.49$. Fig. 9 can again give a visual depiction of what a change of $0.49$ in $\mathcal{L}_{depth}$ might mean for depth prediction. This shows that, starting from a place where $\phi$ works fine, the adversarial attack does move the camera to locations where it struggles.

## 6 DISCUSSION AND LIMITATIONS

When a MDE model fails to give accurate depth for an image, the underlying causes could be multifaceted; it could be the camera viewpoint itself, or the specific combination of objects in a certain lighting condition etc. Consequently, we do not intend our framework to present analysis such as "for *any* scene when yaw & pitch of the camera exceed x° and y°, the MDE model will fail". This is because such values (x°, y°) will very much depend on the particular scene. Hence, the more appropriate way to view our framework is something similar to a testbed which assesses how robust MDE models can be for these 3D scenes if we give the camera full flexibility to move around in the scene. This framework does have its limitations. First, the lack of availability of free-to-use 3D assets that can be converted into an appropriate format. Consequently, our own dataset size is not too big at the moment, focusing only on indoor scenes. Second, while synthetic datasets have been used to train MDE models which then work well for natural images (DA V2 Yang et al. (2024b)), it is not completely clear how well failures on synthetic images translate to failure in real world images.

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

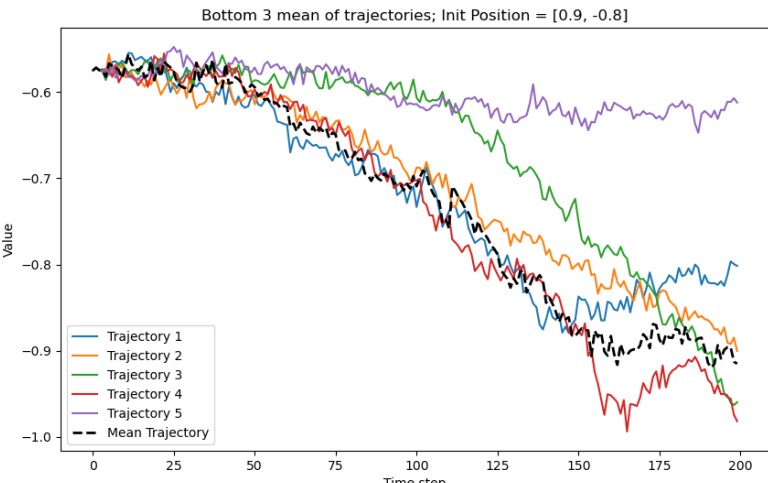

Figure 8: Sample trajectories visualized from the same initialization. Note that while all trajectories lead to a decrease in the negative loss value, each trajectory is different due to the accumulated errors from each render.

## A   3D ASSETS CREATION DETAILS

When working with complex 3D scenes, especially those created in traditional 3D modeling software like Blender, adapting these assets for use with PyTorch3D presents several challenges. A key limitation is that PyTorch3D supports only a single texture per object, which restricts the direct use of assets designed with multiple textures for detailed appearances. Additionally, maintaining the relative positions of objects within a scene requires manually storing positional data and loading each mesh and its corresponding texture, a tedious process that becomes increasingly cumbersome as scene complexity grows.

To address these challenges, we converted publicly available 3D assets of complex scenes—originally created for traditional renderers like Blender[1] into formats compatible with PyTorch3D. This process involves merging all objects in the scene into a single mesh, generating a unified UV map using Blender's Smart UV unwrap, baking all object textures into a single 8K texture map using Blender's Cycles Renderer, and finally loading the processed mesh and texture into PyTorch3D while preserving spatial relationships. By undertaking this conversion, we created a repository of 3D assets that can be rendered directly in PyTorch3D, streamlining workflows for 3D rendering and analysis.

## B   MULTI-RUN STOCHASTICITY

An interesting observation we made during our experiments was the inherent stochasticity in the rendering process. Specifically, we noticed that with PyTorch3D, the exact same viewpoint produces images with small variations on the order of $1e^{-4}$. This generally accumulates over a trajectory leading to slightly different trajectories across multiple runs done from the same initial camera position. This is a known issue with the PyTorch3D renderer nh236 (2023). We also include a visualization of the variance between runs initialized from the same starting point in Figure.8 by running multiple optimizations from the same initialization and plotting the loss over time. The y-axis measures the loss value, and x-axis is the number of iterations. As we can see, the runs deviate with time. Hence, we run our approach five times and pick the best, as described in the main paper (Sec. 3.5). We run our experiments on NVIDIA RTX A4500 GPUs. Overall, obtaining the results of four MDE models on nine assets, as descibed in Sec. 3.5 took about 3 days of runtime on eight NVIDIA RTX A4500 GPUs, each having 20GB RAM.

---

[1]https://mitsuba.readthedocs.io/en/stable/src/gallery.html

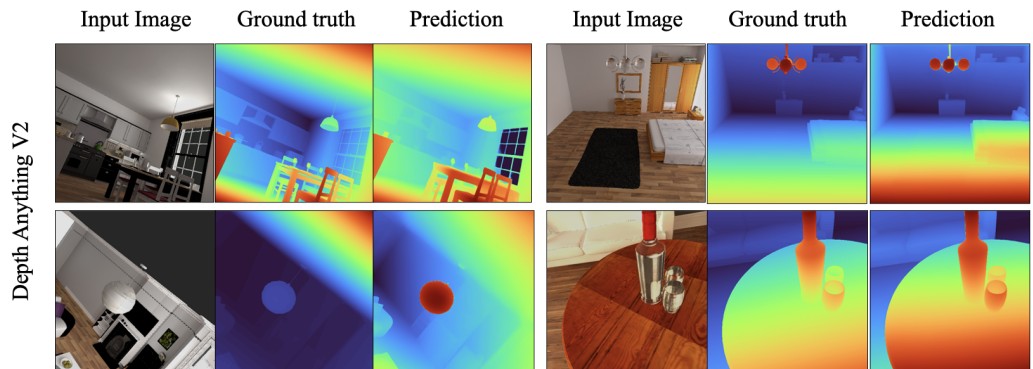

Figure 9: Figure similar to Fig. 6 in main paper, showing how a 3D asset rendered using Blender (top row) looks after being converted and rendered using PyTorch3D (bottom row). The depth maps obtained using Depth Anything-V2 Yang et al. (2024b) for both images are similar, despite the small quality drop in RGB image.

Figure 10: Adversarial viewpoints obtained for Depth Anything V2. Each triplet contains rendered image, ground-truth and predicted depth.

## C EVALUATING SCENES ON NERF/GAUSSIAN SPLATS

While 3DGS and NeRF provide ways to scale the toolbox for more photorealistic large 3D scenes, we found that neither method accurately resolves the scene's depth. In both cases, depth is estimated as part of the rasterization process. For NeRF, depth is determined by estimating the point of termination for each ray. However, this estimation is often inaccurate, as shown in Deng et al. (2024). While this and other works Deng et al. (2024); Rau et al. (2024); Dadon et al. (2023) attempt to improve accuracy by using depth priors, the results still do not match the ground truth generated by rendering a 3D asset. A similar issue arises in 3DGS Xu et al. (2024) and Chung et al. (2023). Additionally, we have observed that large Gaussians in 3DGS can produce inaccurate depth rasters. Neither model generalizes well to viewpoints outside the training distribution, meaning that unbounded camera optimization would require extensive scene coverage. Hence, in this work, we rely on 3D assets that we can be certain will provide accurate depth maps.

702
703
704
705
706
707
708
709
710
711
712
713
714
715
716
717
718
719
720
721
722
723
724
725
726
727
728
729
730
731
732
733
734
735
736
737
738
739
740
741
742
743
744
745
746
747
748
749
750
751
752
753
754
755

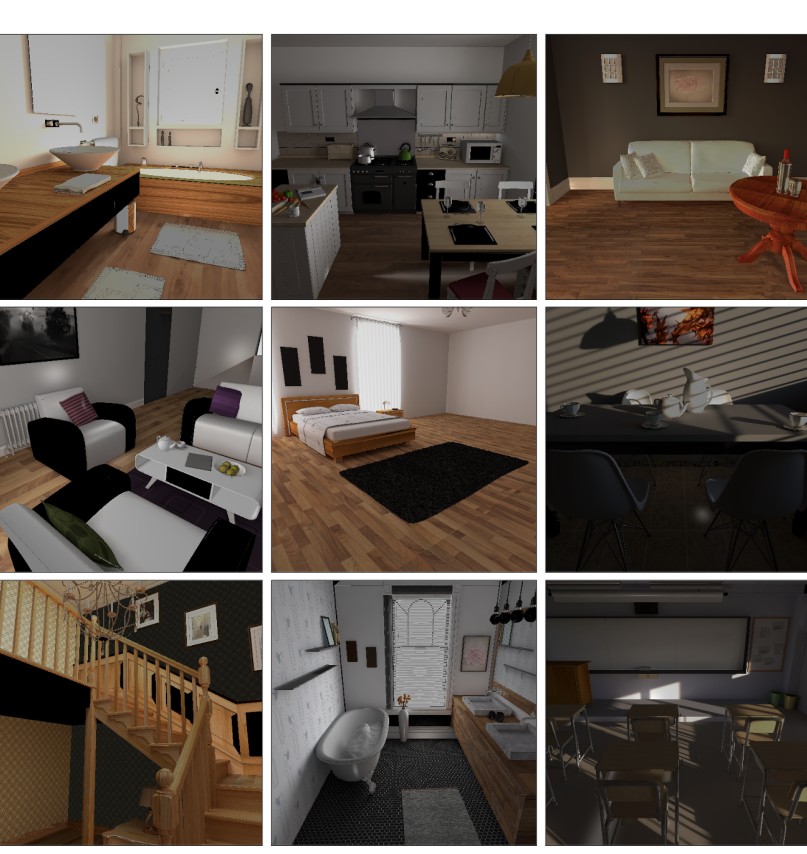

Figure 11: Images from all the nine 3D assets that we use in this work.

