# OpenReview forum: "Breaking Depth Estimation Models with Semantic Adversarial Attacks"
_ICLR.cc/2026/Conference — ICLR 2026 Conference Withdrawn Submission_

### Official Review · Reviewer_ZLJN · 2025-10-30

**Soundness:** 2
**Presentation:** 2
**Contribution:** 2
**Rating:** 2
**Confidence:** 5

**Summary:**

- The paper proposes a semantic adversarial attack on monocular depth estimation (MDE) that optimizes camera extrinsics (rotation and translation) through a differentiable renderer (PyTorch3D) to find viewpoints where MDE models fail. The loss is a perceptual distance between predicted and ground-truth depth (via VGG-16 features), and a geometric penalty deters trivial “shortcuts” like moving the camera into geometry. The authors curate nine indoor 3D assets, convert them for PyTorch3D, and evaluate four models (MiDaS, ZoeDepth, DepthAnything v1/v2). They report consistent failures at adversarial viewpoints and position the work as a robustness benchmark for MDE.

**Strengths:**

- The idea of adversarial viewpoint search exists for recognition; extending it to depth with differentiable rendering and a near-surface penalty is a reasonable, incremental contribution with practical utility as a diagnostic/benchmark.

**Weaknesses:**

- The evaluation setup is quite limited in scope. The experiments rely on just nine indoor 3D scenes, without testing on outdoor environments, complex or dynamic settings, or real-camera data. Additionally, alternatives such as NeRF or 3D Gaussian Splatting scenes (which could offer more realistic diversity) are not explored (though the appendix mentions concerns about their depth quality). As a result, the findings cannot be confidently generalized to real-world scenarios or broader depth estimation challenges.
- The choice of metric raises some concerns. The main optimization objective, Ldepth​, relies on perceptual features extracted from a VGG-16 network that was trained on RGB images, not on depth data. Since depth maps represent geometry rather than texture or color, it’s unclear whether these RGB-trained features meaningfully capture geometric discrepancies. Established depth-specific metrics such as SILog, log-RMSE, scale-invariant RMSE, or edge-aware error would provide a more accurate measure of depth quality. Although the paper includes delta as an evaluation metric, the attack itself is guided by the VGG-based loss, which may push the optimization toward visually different depth maps that look dissimilar in feature space but are not necessarily more geometrically inaccurate.
- The paper lacks proper baseline comparisons for the proposed “semantic search” approach. There are no experiments contrasting the method with simpler or established alternatives, such as random or grid-based viewpoint sampling using comparable computational effort, or gradient-free optimization methods like NES or CMA-ES. Moreover, although the paper cites ViewFool (a prior work on adversarial viewpoint attacks) it does not include a reimplementation or adaptation of that approach for depth estimation as a baseline. Including these comparisons would help clarify the actual advantage of the proposed method over simpler or existing strategies.

**Questions:**

Please check weaknesses.

---

### Official Review · Reviewer_hMGY · 2025-10-30

**Soundness:** 3
**Presentation:** 3
**Contribution:** 2
**Rating:** 4
**Confidence:** 4

**Summary:**

The paper presents a framework for stress-testing monocular depth estimation (MDE) models to find view-dependent failure modes. The authors' method, which they frame as a "semantic adversarial attack," uses a differentiable renderer (PyTorch3D) to find camera poses that maximize the discrepancy between a model's predicted depth and the ground-truth depth .
Given a 3D asset, the 6-DoF camera parameters (rotation and translation) are iteratively optimized via gradient ascent to increase a perceptual loss. A key aspect of this method is that the ground-truth depth map changes at each step along with the camera pose, a distinction from many existing adversarial attacks 3. To prevent degenerate solutions, the framework includes a penalty term ($\mathcal{L}_{dist}$) that repels the camera from mesh surfaces, preventing "unnatural" views like clipping inside objects

**Strengths:**

- The paper offers a tool for exploring view‑dependent failure cases in depth networks.
- The work is well presented and the robustness of depth networks is an important topic.
- The conversion of nine 3D assets into a PyTorch3D-compatible format is a useful, although engineering-focused, contribution that enables this specific type of differentiable, non-static analysis

**Weaknesses:**

## Major

1. **Lack of External Validity / Downstream Impact:** The paper's analysis stops at reporting pixel-level metrics ($\mathcal{L}_{depth}$ and $\delta_1$). While it shows that predictions look different, it fails to demonstrate why this matters. The core unaddressed question is: **do these "adversarial viewpoints" cause failures in downstream tasks?** I understand the paper has been framed as a testbed, yet, an experiment showing that -for example- a robot's navigation or obstacle avoidance algorithm fails, or that a SLAM system's pose estimation drifts, would make the work significantly more impactful
2. **Comparative Baseline:** The framework is presented in isolation. There is no comparison to other methods for finding failure cases. For instance, how does this prposed optimization framework compare to a simpler naive stochastic (random) search for bad camera poses? A comparaison against other optimization-based viewpoint attacks cited in the related work could have been useful to understand/evaluate the impact/effectiveness

## Minor
* **Synthetic-to-Real Gap:** The paper relies entirely on synthetic data. While the authors acknowledge this as a limitation , there is no discussion on how these failures might translate to real-world scenarios, which involve sensor noise, motion blur, and dynamic objects.

**Questions:**

The paper has a well-framed core idea. However, to be a convincing, it must demonstrate (a) why its findings matter for downstream tasks and (b) how its effectiveness compares to simpler baselines.

---

### Official Review · Reviewer_LJK1 · 2025-10-31

**Soundness:** 3
**Presentation:** 3
**Contribution:** 3
**Rating:** 4
**Confidence:** 4

**Summary:**

The paper proposes a semantic adversarial attack on monocular depth estimation (MDE) models by optimizing camera extrinsics (rotation and translation) to maximize prediction error. Using differentiable rendering (PyTorch3D) and a perceptual loss, the authors search for failure-inducing viewpoints in indoor 3D scenes. They curate 9 complex indoor 3D assets compatible with PyTorch3D, evaluate 4 SOTA MDE models (MiDaS, ZoeDepth, Depth Anything V1/V2), and show that all models fail significantly on optimized adversarial viewpoints. The work introduces a new robustness benchmark for MDE under viewpoint variation.

**Strengths:**

1.	Introduces camera pose–based adversarial attacks for MDE (Sec. 3), moving beyond pixel-level perturbations studied in prior works such as ViewFool (Dong et al., 2022).
2.	The differentiable rendering pipeline and optimization formulation are rigorously explained and well-motivated.
3.	The creation of PyTorch3D-compatible 3D assets fills an existing gap in robustness research for depth models.
4.	Evaluations cover multiple models, assets, and metrics (Table 1–3), providing both quantitative and qualitative evidence (Fig. 6–7, 10).

**Weaknesses:**

1.	All experiments rely on synthetic indoor scenes; real-world RGB-D data validation such as  ( NYU V2, ScanNet) is missing (acknowledged in Sec. 6).
2.	Only nine indoor assets (Fig. 11) restrict generalization; inclusion of outdoor or dynamic scenes could improve coverage.
3.	While Fig. 7 visualizes Ldepth, explicit thresholds linking its magnitude to failure severity are absent.
4.	No direct quantitative comparison to other semantic attacks (e.g., MeshAdv (Xiao et al., 2019), ViewFool) is included, leaving the gain over prior art somewhat implicit.
5.	Although Fig. 6–10 visualize cases, the paper lacks deeper analysis of why specific viewpoints cause breakdowns for example (occlusion, object proximity, or lighting).

**Questions:**

1. How does the framework scale computationally? Can you provide a runtime breakdown for rendering, optimization, and inference?
2. How sensitive are the attack results to initial camera poses or optimizer hyperparameters?
3. Could a coarse-to-fine optimization scheme (lower-resolution renderings in early iterations) preserve attack strength while reducing cost?
4. To what extent does DepthAnything V2’s training on synthetic data (Hypersim) contribute to its improved robustness (Table 2)?
5. Would the framework support non-differentiable renderers or hybrid scenes in future work?
6. Add a summary plot showing δ₁ degradation per model and asset (derived from Table 3) for easier cross-model comparison.
7. Conduct at least one real-world validation test, even if limited ( on NYU V2 RGB-D scenes).
8. Discuss potential differences like ( pose augmentation, multi-view consistency) that could mitigate viewpoint attacks.

---

### Official Review · Reviewer_X1N6 · 2025-11-03

**Soundness:** 2
**Presentation:** 3
**Contribution:** 3
**Rating:** 4
**Confidence:** 3

**Summary:**

This paper proposes a semantic adversarial attack against a new application domain, namely, monocular depth estimation. Specifically, the paper aims to demonstrate the vulnerability of existing depth estimation methods to changes in camera viewpoints. To find the adversarial camera position that causes the depth estimation to fail, the authors design a semantic adversarial attack using a differentiable renderer. The authors describe the overall attack architecture, including the target loss function, handling changes in ground truth depth, and penalties for the attack taking a shortcut. Through experiments on nine different assets, the paper presents the example attack results and the quantified discrepancy introduced by the attack.

**Strengths:**

1. To the best of my knowledge, the proposed method is novel.
2. The paper has a good motivation. In particular, the paper aims to find the limitations of cutting-edge depth estimation models. This is not an easy challenge, and identifying the limitations of existing models should be encouraged.
3. Semantic adversarial attack is an important research topic in machine learning, and it is a valuable method to demonstrate the existing vulnerability of the target model even without adding visible adversarial noise. Identifying a new application domain for semantic attacks is a good contribution to machine learning research.

**Weaknesses:**

1. The number of samples is not sufficient to demonstrate the effectiveness of the proposed method.
2. The paper does not present the baseline of the quantified discrepancy. For example, both discrepancy metrics can be measured on the benign (before-attack) samples. Without the baseline, it is unclear how much adversarial effect the proposed method introduced.
3. The adversarial attack takes the gradient of the composition, i.e., the renderer followed by the depth estimation. It is still possible that the adversarial attack influenced the renderer’s behavior. For example, the adversarial camera position could be the camera position where the differentiable renderer is not accurate in rendering.
4. Similarly, the target loss computation (i.e., LPIPS) also involves another network, namely VGG-16. While the presented results all showed visible differences in the depth map, it is still possible that some imperceivable depth map change could induce a considerable loss.

**Questions:**

1. There are not many examples that satisfy the conditions that the author mentioned. However, a sample size of nine is too small to generalize the findings. Please find a way to overcome this problem.
2. To demonstrate the effectiveness of the proposed method, it is essential to provide a baseline to compare. Please provide the pre-attack discrepancies so that readers can see the increase in discrepancies after the attack.
3. As mentioned in the Weaknesses, it would be better to use two renderers: one for computing the adversarial camera position and the other (e.g., Blender) for rendering the computed camera position. If the newly rendered image is still adversarial, we can attribute the adversarial effect to the camera position, rather than the renderer’s failure.

---

### Note · Authors · 2025-11-12

I have read and agree with the venue's withdrawal policy on behalf of myself and my co-authors.